# Liquid Biopsies Poorly miRror Renal Ischemia-Reperfusion Injury

**DOI:** 10.3390/ncrna9020024

**Published:** 2023-04-01

**Authors:** Adaysha C. Williams, Vaishali Singh, Pengyuan Liu, Alison J. Kriegel

**Affiliations:** 1Department of Physiology, Medical College of Wisconsin, Milwaukee, WI 53226, USA; 2Department of Pediatrics, Medical College of Wisconsin, Milwaukee, WI 53226, USA; 3Cardiovascular Center, Medical College of Wisconsin, Milwaukee, WI 53226, USA

**Keywords:** microRNAs, renal ischemia, liquid biopsy

## Abstract

Acute kidney injury (AKI) is the rapid reduction in renal function. It is often difficult to detect at an early stage. Biofluid microRNAs (miRs) have been proposed as novel biomarkers due to their regulatory role in renal pathophysiology. The goal of this study was to determine the overlap in AKI miRNA profiles in the renal cortex, urine, and plasma samples collected from a rat model of ischemia-reperfusion (IR)-induced AKI. Bilateral renal ischemia was induced by clamping the renal pedicles for 30 min, followed by reperfusion. Urine was then collected over 24 h, followed by terminal blood and tissue collection for small RNA profiling. Differentially expressed (IR vs. sham) miRs within the urine and renal cortex sample types demonstrated a strong correlation in normalized abundance regardless of injury (IR and sham: R^2^ = 0.8710 and 0.9716, respectively). Relatively few miRs were differentially expressed in multiple samples. Further, there were no differentially expressed miRs with clinically relevant sequence conservation common between renal cortex and urine samples. This project highlights the need for a comprehensive analysis of potential miR biomarkers, including analysis of pathological tissues and biofluids, with the goal of identifying the cellular origin of altered miRs. Analysis at earlier timepoints is needed to further evaluate clinical potential.

## 1. Introduction

Acute kidney injury (AKI) is a rapid reduction in renal function that occurs over the course of hours to days. In recent years, several studies have emerged that have characterized the differential regulation of microRNAs (miRs) in response to renal ischemia-reperfusion injury (IRI), a common cause of AKI [1,2,3,4,5,6]. Alterations in miRs have also been explored within bodily fluids with the goal of identifying biomarkers of AKI and other diseases. It remains unclear whether pathologically altered renal miRs are released into the urine or circulation, where they could serve as biomarkers or even potentially mediate target regulation in distant cell types and organs.

Thousands of publications report the potential utility of miR biomarkers in disease diagnosis or progression. Extracellular membranes protect miRs from rapid degradation by RNases within the blood and other body fluids, making analysis of these easily accessible and relatively stable sample types attractive for biomarker identification [7,8]. These studies often nominate predictive miR biomarkers and propose a link to related disease processes simply by referencing published findings on the miR in cells or solid tissues. Given the difficulty of obtaining renal tissues at the time of AKI events in human subjects, this study utilized a rat model of renal IR to optimize the opportunity to capture IR-induced miR changes in urine and plasma “liquid” biopsies and compare them to changes in the renal cortex under controlled experimental conditions.

The renal cortex makes up the majority of the renal mass and is highly susceptible to damage from ischemia due to its high aerobic metabolism [9,10]. We hypothesized that, given the rapid vesicular shedding by proximal tubules in response to ischemic stress [11], urinary and plasma miR profiles would correlate with changes in the renal cortex. The goals of this study were two-fold. The first goal was to identify miRs that were significantly altered in abundance in the renal cortex, urine, and plasma resulting from 30 min of bilateral ischemia followed by 24 h of renal reperfusion (IR). The second goal of this study was to determine if changes in the miR abundance in the urine or plasma could serve as a true “liquid biomarker”, directly reflecting miR changes in a damaged organ. In this study, the cause of the renal ischemia was complete physical occlusion of the renal artery and vein, allowing us to attribute subsequent expression changes to this isolated initiating event. The controlled study design allowed differences in urine and plasma miR abundance to be directly attributed to either release by the damaged kidney or as part of a systemic response to IR injury.

## 2. Results

### 2.1. Reduced Renal Clearance in This IR Model

Biochemical analysis was performed on serum samples collected 24 h after renal reperfusion. Basic parameters including body weight, kidney mass, and urine volume are presented in Appendix A. As anticipated, IR animals exhibited significantly elevated serum urea and creatinine, indicative of reduced renal function (Table 1). Additionally, phosphorous was elevated by IR, while serum sodium and bicarbonate were reduced (Table 1). Laboratory values also suggest impairment in liver function secondary to renal IR injury.

### 2.2. Small-RNA Sequencing Results

In sum, small RNA-sequencing and subsequent alignment of the three sample types identified a total of 528 unique miRs in our dataset. Mapping rate, miR counts, and results of statistical analysis are presented in Appendix A. Only differentially expressed (DE) miRs with and adjusted *p* value (adj. *p* value) < 0.05 IR vs. sham within any sample type were included in subsequent analysis (Figure 1). Notably, IR resulted in 161 unique DE miRs in the remaining three sample types (cortex, plasma, and urine). Of the 161 DE miRs, 128 were unique to a single sample type, while 33 were DE in at least two sample types (Figure 1). Subsequent filtering steps identified 98 unique DE miRs with 100% seed homology to human, mouse, or rat across all 3 samples, 19 of which have been previously reported in studies of IR. None of the 19 miRs were DE within multiple sample types, and the majority were DE in cortex tissue (Figure 1).

### 2.3. Overall Correlation of Abundance between Sample Types

Pearson’s correlation analysis was performed to determine if there were correlations between normalized miR abundance (counts per million; CPM) in liquid sample types and/or cortex tissues collected within each experimental group (Figure 2). miRs included in this analysis were identified as DE with IR in each of the sample types within the indicated comparison. This analysis failed to identify correlations between cortex and plasma samples in sham or IR groups (Figure 2A,B, respectively). Similarly, there was no correlation between the abundance of urine and plasma samples in sham and IR samples (Figure 2C,D, respectively). There was, however, a strong positive correlation in both sham and IR renal cortex and urine samples: R^2^ = 09716 and 0.8710, respectively (Figure 2E,F), indicating the overall abundance of urinary miRs correlates with the abundance of miRs in cortex tissue.

### 2.4. Analysis by Nomenclature

The directionality of IR-related changes in the 161 unique DE miRs identified in the cortex, urine, and plasma samples are depicted in Figure 3A. Of the 161 unique DE miRs in our dataset, only 98 share 100% seed sequence homology with human, mouse, or rat species (Figure 3B). The top ten, or fewer, DE miRs with the largest-fold change following IR in the renal cortex, plasma, and urine are depicted in Figure 3B–D, respectively. Changes in DE miRs that were differentially expressed in either cortex and plasma or urine and plasma samples are presented in Appendix A, respectively.

### 2.5. Homology and Abundance

We then screened DE miRs in the IR vs. sham comparison to identify miRs with 100% shared seed region homology between rat, mouse, and human, finding 98 miRs in total. Of those, 84 were only DE in 1 sample type (Figure 1, Appendix A). An additional six were DE in both plasma and urine samples (Figure 1, Appendix A), and the remaining eight were DE in both plasma and cortex samples (Figure 1, Appendix A). Those miRs DE in both plasma and urine samples and plasma and cortex samples are shown by log2 fold-change in Figure 4A,B. We then explored the abundance of miRs between rats, mice, and humans that were identified as DE in urine (Figure 4C) and plasma (Figure 4D) samples. The number of miRs increased in abundance in IR samples was much higher in plasma than in urine samples. Though several DE miRs were detected at a relatively high abundance, there were many with an overall low mean abundance.

### 2.6. Prior Identification in Related Studies

After identifying the 98 DE miRs with 100% shared seed region homology between rats, mice, and humans among all samples analyzed, an extensive literature review was performed to determine if any of the miRs we identified had been previously reported in any published studies on renal ischemia, AKI or IR pathology performed on samples from rats, mice, or humans. This analysis yielded 19 miRNAs (Figure 1). Of these 19 miRs, 5 were detected in plasma, 11 were detected in cortex and 3 were detected in urine in this study (presented with other studies cited in (Table 2).

## 3. Discussion

There has been tremendous interest in identifying novel biomarkers of renal injury in easily accessible biofluids [31,39,41,47]. The potential for miRs to serve as sensitive biomarkers of kidney-specific pathology in a clinical setting necessitates that changes in biofluid profiles reflect, or are otherwise specifically attributable to, renal pathology [26,29,33,48,49,50,51] and ultimately superior to established biomarkers of renal dysfunction (e.g., BUN, SCr, KIM-1, etc.) [23,31,39,47]. Relatively few studies have included a comprehensive analysis of multiple sample types in addition to kidney tissue [24,25,26,39,43,44,47,52,53,54]. In this study, the renal damage was reproducibly isolated to a fixed period of IR injury (IRI), resulting in changes to the abundance of 161 annotated miRs over the three sample types studied (i.e., cortex, plasma, and urine). Of those miRs identified as differentially expressed, only 98 share human, mouse, and rat seed sequence homology. The literature review revealed that only 19 miRs have previously been reported in rodent or human studies of AKI, further highlighting the challenges in using miRs to accurately predict or diagnose AKI.

The differences in miR expression profiles may be model-dependent and might not represent clinical findings, but many miRs are conserved across mammalian species suggesting many identified in a rat model could have relevance to mouse models and research in samples from human subjects. An analysis of results from this study was designed to identify DE miRs that may have biological specificity and relevance to human AKI pathology. A common approach for identifying potential miR biomarkers is to identify miRs that correlate with clinical markers of renal damage, leading to a similar conclusion—that miRs, especially after injury, tend to deregulate common pathways related to cell proliferation, death and inflammation [31,39,48]. To date, most miRs involved in renal IR/AKI, in human and animal models, have established involvement with other non-specific disease processes such as fibrosis, repair, and even oxidative stress [31,39,48,49,50,51,55]. In humans who may have complex comorbidities and concurrent chronic diseases, it is essential that miR-based biomarkers would be indicative of only AKI.

A change in the abundance of miRs in biofluids in IR could occur for many reasons. An increase in biofluid miR abundance could reflect the increased pathological expression of that miR within the kidney. Alternatively, elevated miR abundance in biofluids could result from miR release from tissue damage, resulting in subsequently reduced tissue abundance. In a study evaluating miRs as biomarkers of kidney segment-specific injury induced by doxorubicin, Church et al. (2014) attributed the decrease in renal miR–34c–3p to leakage into the urine, which contributed to elevations of urinary miR–34c–3p there [53]. Many other scenarios are also possible. In this study, the highly controlled protocol used to induce isolated renal injury, allowed us to identify alterations in bioliquid miRs resulting directly from either ischemic injury to the kidney or systemic consequences secondary to IR. Systemic changes must be considered since we find several biochemical indices of disrupted systemic homeostasis in serum samples collected 24 h post-IR surgery (Table 1). This study identified five conserved miRs that were increased by IR in both the urine and plasma samples (Appendix A). Additionally, six miRs showed reciprocal relationships of decreased abundance in the cortex and increased abundance in the plasma (Appendix A). No overlap of conserved DE miRs in urine and cortex samples was found, despite a strong correlation between the abundance of DE cortex and urine miRs (Figure 2E,F). While this data could be interpreted to indicate that cortical miRs are released into the circulation, and not the urine, following IR, caution is needed because of the possibility that circulating or urinary miRs were produced by cells outside of the kidney and for reasons detailed in subsequent sections.

### 3.1. Reflection of Organ-Specific miR Changes in Biofluids

Many miRs—including those found in our dataset—have a reported role in cellular pathways that are common to organs throughout the body. Several studies have reported differentially expressed miRs following IR injury; some reports align and others conflict [26,29,33,48,49,50,51]. For example, Godwin et al. (2010) reported an increase in miR–20a, miR–21, and miR–199a in kidney tissue following 30 minutes of unilateral renal ischemia [18]. The present study also identified an increase in miR–20a and miR–21 (–5p and –3p) in the cortex, but no change in miR–199a. Another hypoxia/reoxygenation study, also induced by similar dual renal pedicle clamping, reported a significant reduction in the abundance of miR–20a in the renal cortex following IR [33]. This is just one example of how differences in models, methods, and sample types may result in a convoluted understanding of the role of specific miRs in disease processes and as disease biomarkers, even in tissues. To ultimately determine if miRs can reliably serve as liquid biomarkers, AKI induction, time to injury, methodology (broad or targeted assays) and sample types are all variables that must be considered.

An important criterion for biomarker selection is the ability to determine disease specificity, and in this case, the cellular origin of DE miRs. The identification of kidney injury marker 1 (KIM-1) was extremely valuable as an organ-specific biomarker [56]. To date, a kidney-specific miR has not been identified. There are two general approaches that have been taken to identify potential miR biomarkers: (1) determine if changes in organ-specific miRs are reflected in biofluids, and (2) if changes in miRs within biofluids are aligned/associated with established biomarkers of disease. The present study aimed to do both, using an unbiased transcriptomic-based approach. Importantly, the relative abundance of each miR is available through this approach, in addition to fold-change in abundance.

As an example of the first approach—changes in organ-specific miR reflected in biofluids—three separate studies focused on methods of renal IR in rodent or cell culture models similarly observed elevated miR–18a levels in the renal cells following renal IR [28,32,57]. Despite increased abundance in the cortex in this study, miR–18a was not significantly changed in urine or plasma samples from IR rats. The results of this study suggest that biofluids sampled were unable to detect this well-established specific renal tissue-level change. Critical comprehensive review of literature conducted throughout this study revealed themes of non-specificity of miR abundance changes across different sample types and disease etiologies, which confounds the assessment of miRs as biomarkers of renal disease based on the result of any single study.

### 3.2. Alignment of Biofluid miRs with Established Biomarkers

Serum creatinine and blood urea nitrogen (SCr and BUN) are standard clinical biomarkers used to assess kidney function [23,31,39,47]. While both are indirect measures of renal disease, clinicians and scientists appreciate the need for accurate biomarkers of AKI that could impact decisions for patient care at an earlier stage of disease (i.e., before renal function declines enough to result in elevated SCr and BUN). However, the incorporation of these established biomarkers, in both human and animal studies, is an important first step in evaluating the ability of miRs as biomarkers that match their diagnostic value [23,24,25,26,31,39,43,44,47,52]. In most retrospective human studies, the incorporation of healthy donor controls and even tissue acquisition is important in identifying specific renal biomarkers. One of the challenges with this approach is that miRs can be differentially expressed between subjects for reasons unrelated to the disease of interest (i.e., comorbidities, viral infection, other illnesses, etc.) [58,59,60]. A study by Bellinger et al. (2014), utilized qRT-PCR to identify several DE miRs, including miR–877*(–3p), in both the kidney and plasma samples following bilateral renal ischemia in mice [46]. The miR–887* did not correlate with creatinine in their analysis and was not detected in analysis in this study, though miR–887(–5p) was increased in IR urine at a very low abundance (Figure 4C). Alternatively, Glineur et al. (2018) found increased urinary miR–34c outperformed KIM-1 (a urinary biomarker) as a positive correlative with cisplatin-induced AKI in rats [41]. They also reported that the enriched pathways for miR–34c were in non-renal related diseases (testis apoptosis and gastrointestinal tract) [41]. Urinary miR–34c has been reported to be increased in response to cisplatin treatment [41,61], and it has been reported in renal injury resulting from numerous nephrotoxic agents including doxorubicin and N-phenylanthranilic acid [61]. The study by Chorley et al. (2021), rigorously and elegantly aimed to link urinary miR changes with nephrotoxicant-induced nephron segment-specific changes; however, the overarching problem remains—many of these nephrotoxicants also are also toxic to non-renal cells and tissues when administered systemically. It cannot be assumed DE urinary miRs came from the kidney even if the kidney is damaged. The lack of overlap in DE genes in the kidney-specific insult of IR highlights the need for further studies to determine if it will ultimately be possible to attribute DE miRs from biofluids to specific disease processes or even cell/tissue types [39,48].

Despite a focus on miRs with seed region conservation in rats, mice, and humans, the present study did not identify much alignment with reports from similar liquid biomarker studies conducted in rodent or human samples. An example of this is a study that looked at miR profiles from plasma exosomes collected from patients with delayed graft function, a clinical manifestation of IR, that resulted in increases in miR–26a, –145, –150, –199a–3p, –486, –20a, –98, –21, and let–7a [42]. None of these miRs were identified as increased in the plasma after IR in our dataset. The authors also noted a limited correlation with clinical parameters of renal function (SCr, BUN, or uric acid) [42]. In another study of urine collected from patients with AKI (ischemic or septic), miR–21 was identified as increased when compared to that of healthy volunteers, but this did not match their findings in an animal model of gentamicin-induced AKI where urinary miR–21 abundance was decreased [48]. Importantly, miR–21 is also reported in numerous pathological processes including cancer [62,63,64,65] and fibrosis [65,66,67]. Other miRs are also implicated as biomarkers in numerous disease processes. This further highlights the need for investigation of miRs within and between multiple sample types and disease conditions to work toward validation of miRs from biofluids as disease-specific biomarkers.

### 3.3. Importance of Study Design

It is important to limit biases by the implementation of research designs that prioritize comprehensive sample collection and analyses via sequencing; this is less likely to limit the number of miRs that fail to be identified and increase the chances of novelty. Subsequently, it provides relative abundance levels, rather than simply reporting fold change. This information provides nuance to the complexity of both health and disease. Currently, there are a limited number of studies that test multiple sample types or address reproducibility before claiming potential miRs as biomarkers. The studies presented provided an unbiased dataset and a targeted approach to assess miR abundance in three sample types, two of which are measured readily in the clinic (plasma and urine). While pathway analysis and gene target predictions are useful, it is important to verify that the miR is reflective of changes occurring in an injury model. More studies are needed to address miR origin and disease specificity to verify that miR changes resulting from AKI are reflected in circulating miR levels. This study also highlights the need for more comprehensively planned experiments that assess miR function, origin, and relevance to human AKI.

### 3.4. Limitations

As with other biomarker studies, this study has inherent limitations. Though the controlled nature of this study allowed us to determine DE miRs detected in each sample type are likely a result of bilateral IR, the cellular origin of miRs altered in urine and plasma cannot be determined. This is a limitation for the field as it has not been possible to distinguish the origin of miRs based on circulating levels [39,48,68]. Another consideration of the results from this study is that we did not specifically isolate and analyze extracellular vesicles of a particular size for RNA profiling, rather we profiled the sum of extracellular vesicles of all sizes within urine and plasma samples. This may explain some of the differences between our results and other studies of IR models where exosomes were specifically studied. Finally, we only profiled renal cortex tissue and are unable to evaluate IR-induced expression changes in other kidney regions.

## 4. Materials and Methods

### 4.1. Ischemia-Reperfusion (IR) Model

Animal work was approved by the Medical College of Wisconsin’s Institutional Animal Care and Use Committee (IACUC). Young adult (10-week-old) male Sprague Dawley (SD) rats were anesthetized by i.m. injection of a ketamine (50 mg/kg)/xylazine (8 mg/kg)/acepromazine (5 mg/kg) mixture and maintained at 37 °C throughout surgery and recovery from anesthesia. A midline laparotomy was performed followed by either 30 min of bilateral ischemia by clamping of the renal artery and vein at the renal pedicle (*n* = 7), or sham surgery (*n* = 5) consisting of manipulation of the renal artery and vein. Clamps were removed from IR animals and observed for “pinkening”, indicative of restored blood flow, before the abdominal incisions were closed. Following recovery from anesthesia, animals underwent urine collection in metabolic caging for remainder of the 24 h period (approximately 22 h) following IR before undergoing anesthesia for tissue collection as described below.

Blood was collected by cardiac puncture. For each animal, a total of 1.5 mL of blood was collected for serum analysis and 3 mL of blood was collected into an EDTA-laced tube at 4 °C for plasma isolation. For serum collection, blood was incubated at room temperature for 20 min before centrifugation at 1000× *g* for 10 min at 4 °C. Serum supernatant was promptly shipped to Marshfield Labs (Marshfield, WI 54449) for analysis. Blood collected into EDTA was also centrifuged at 1000× *g* for 10 min at 4 °C.

The plasma supernatant was aliquoted into 3 tubes of 200 μL and frozen at −80 °C until RNA extraction. Kidneys were rapidly isolated, weighed. and bisected. Cortex tissue from the left kidney was isolated and snap frozen in liquid nitrogen. Tissues were stored at −80 °C until RNA extraction.

### 4.2. RNA Extraction and MiR Library Prep

The entirety of the urine collected in the metabolic caging during the 24 h following renal reperfusion was collected. Urine was filtered through a 100 μm filter and then the filtrate was centrifuged at 2800× *g* to remove cellular debris, crystals, and visible bacteria, which was confirmed by microscopy. A total of 1.5 mL of urine was centrifuged at 100,000× *g* for 2 h at 4 °C to pellet miRNA-containing small membrane-bound vesicles and exosomes. The supernatant was removed leaving approximately 50 μL of pellet for subsequent RNA isolation.

Total RNA was isolated from 600 μL of plasma (200 μL × 3 tubes) using the miRNeasy Serum/Plasma Kit, following manufacturer instructions. An exogenous spike was not used in this analysis to ensure that the distribution of sequencing results was not skewed if endogenous miRs were extremely low in plasma-derived samples.

Snap-frozen cortex tissues were cryogenically pulverized from RNA isolation using a standard TRIzol (Invitrogen Inc., Waltham, MA 02451)/chloroform isolation, as previously described [69]. Approximately 50 μL of pelleted urine material was put through RNA isolation using this approach [69] and resuspended in 10 uL of RNase-free water.

### 4.3. Library Construction, Sequencing, and Analysis

miRNA libraries were constructed using the Small RNA Library Kit (Illumina Inc., San Diego, CA 92122) as previously described for cortex tissue and with minor modifications for plasma and urine sample RNA [52]. For plasma samples, 5 μL of the 25 μL of eluted RNA was advanced to library construction. For urine samples, 5 μL of 10 μL of resuspended RNA was advanced to library construction. Tissue (cortex) libraries were generated with 11 PCR cycles, while libraries constructed from urine and plasma samples were amplified using 21 cycles. Cluster generation, sequencing, and analysis were performed at the MCW GSPMC Sequencing Core as previously described [70]. The resulting miR reads from IR and sham groups were analyzed for statistical differences within sample type (i.e., IR vs. sham) using the DeSeq2 approach [71]. A Benjamini–Hochberg (BH)-adj. *p* value < 0.05 in IR vs. sham comparison was considered a statistically significant difference.

### 4.4. Post-Hoc Analysis of miR Profile Data

The miRBase database and NCBI blast tools were used to filter only those miRs with seed sequences common in rat, mouse, and human genomes. Differentially expressed miRNAs in each sample type were filtered for potential relevance to IR pathology in other rodent models or human subjects using additional criteria (Figure 1). Homology to humans, mice, or rats was determined by using miRBase and NCBI Blast tools. For those with shared homology, the literature search was conducted via PubMed using terms including: “miR# and ischemia-reperfusion”, “miR# and ischemia”, “miR# and kidney”, or “miR# and kidney injury” to identify miRs previously determined to be altered in kidney damage/injury or kidney ischemia-reperfusion.

Correlation analyses of DE miRs in each sample type were performed in GraphPad Prism. A list of all DE miRs resulting from IR in either sample type was generated to include the mean normalized miR abundance levels from each sample type and experimental group. These abundances were then plotted against each other using a non-linear line (log-log X and Y both log) within each experimental group (e.g., sham or IR). The R^2^ for each comparison was calculated based on the goodness of fit to data points.

## 5. Conclusions

This study was unable to identify DE miRs from biofluids that had both highly conserved seed regions (between mice, rats, and humans) and could be directly linked to changes in cortex abundance. Urine samples contained no DE miRs that were also DE in the renal cortex, while plasma samples contained only eight that were also DE in renal cortex. Because of the limited identification of miRs as potential liquid biomarkers in this highly controlled study, it remains unclear if miRs can be accurately measured in plasma or urine samples as direct markers of kidney injury rather than serving as a snapshot of complex changes in the multi-organ cellular environment at that time. It further highlights discrepancies in miR reporting as biomarkers with reliable clinical translatability [20]. Not only common to animal studies, the theme of determining miR origin still presents an issue—how will clinicians determine primary organ insult? The problem therein becomes if these pathways are activated anywhere else in the body, the ability for a miR to serve as a biomarker becomes muddled.

## Figures and Tables

**Figure 1 ncrna-09-00024-f001:**
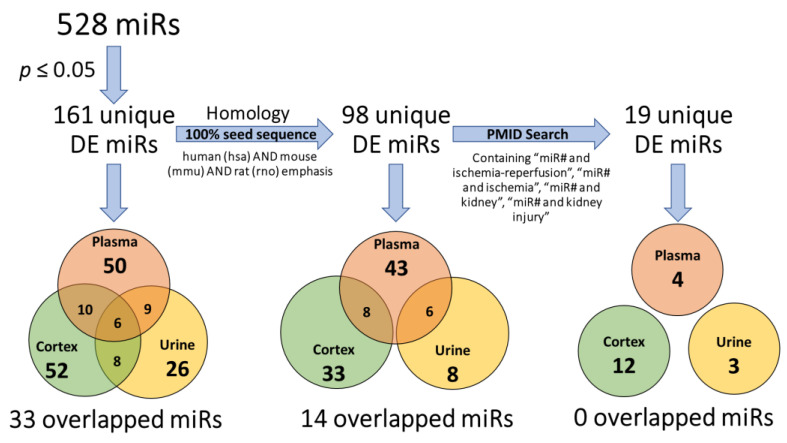
miRNA selection flow chart (left to right). Three criteria thresholds were set for the 528 assayed miRs to filter conserved DE miRs relevant to ischemia-reperfusion injury studies. (1) All miRs with a *p*-value > 0.05 were excluded; 161 unique DE miRs were significantly and differentially expressed following IRI. (2) Based on their seed sequence, only 98 unique DE miRs 100% (seed sequence and polarity) matched to human, mouse, and rat genomes. In the final filter, PubMed was searched for studies involving these 98 miRs and IR injury, identifying 19 examples.

**Figure 2 ncrna-09-00024-f002:**
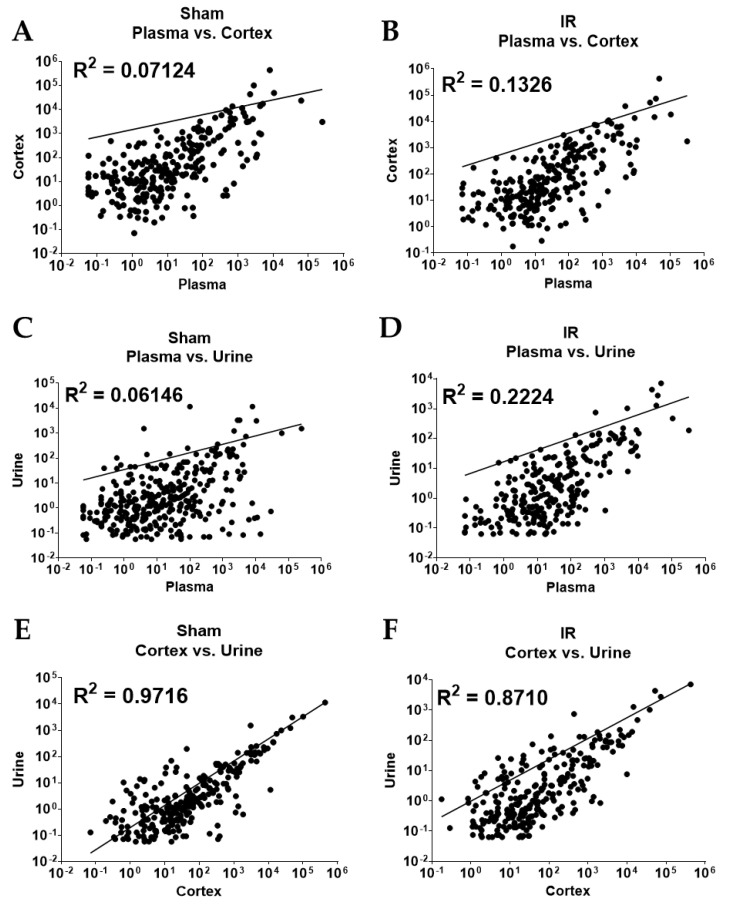
Correlation of miRNA abundance in CPM in DE miRs that appear in two samples from ischemia- reperfusion injured rats compared to sham-operated rats. miRs present in two sample types from both sham and IR groups were analyzed for correlation (R^2^), respectively. (**A**,**B**) Plasma vs. Cortex; (**C**,**D**) Plasma vs. Urine; and (**E**,**F**) Cortex vs. Urine.

**Figure 3 ncrna-09-00024-f003:**
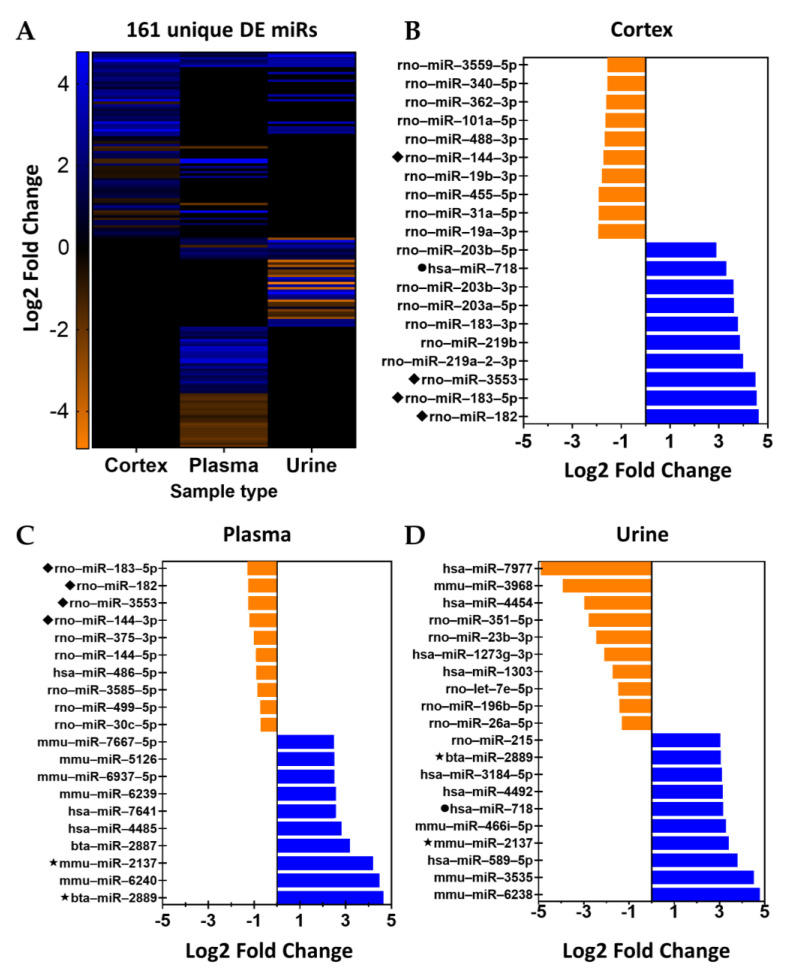
Significantly and differentially expressed miRNAs in samples from ischemia-reperfusion injured rats compared to sham-operated rats. (**A**) Heat map showing 161 DE (adj. *p* value < 0.05) miRs following 30 min of ischemia and 24 h of reperfusion compared to sham-operated controls; each row represents an individual miR and its log2fold-change in abundance within cortex, plasma, and/or urine samples. Ten miRs with the highest log2fold-change increase or decrease in abundance in IR vs. Sham comparison within (**B**) cortex, (**C**) plasma, and (**D**) urine samples. Orange = decreased and blue = increased. ◆ = miR present in cortex and plasma samples; ★ = miR present in cortex and urine samples; and ● = miR present in plasma and urine samples.

**Figure 4 ncrna-09-00024-f004:**
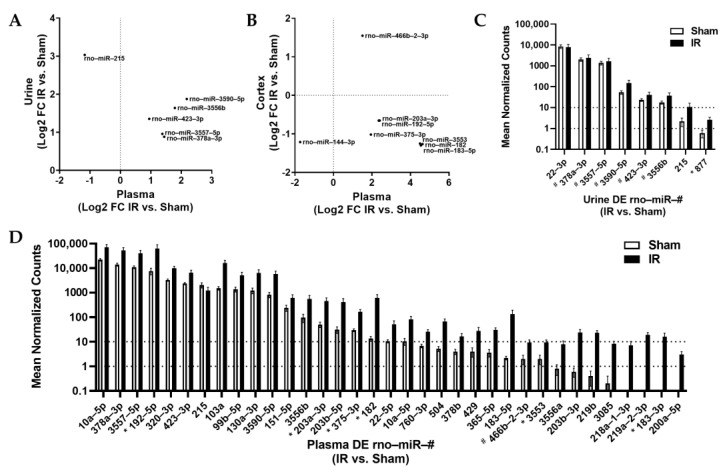
Analysis of miRs with 100% seed region homology (conserved) between rats, mice, and humans. Several DE miRs were identified in both (**A**) urine and plasma samples, and (**B**) cortex and plasma samples, with log2 fold-change IR vs. sham plotted for each miR and sample type. Points represent the mean value for each miR, within the study group. The relative abundance of conserved miRs that were significantly increased in IR vs. sham comparisons for (**C**) urine and (**D**) plasma samples. Mean ± SEM. For (**C**), ^#^ designates increased in IR plasma and * designates decreased in IR plasma. For (**D**), ^#^ designates increased in IR cortex and * designates decreased in IR cortex. All miRs presented had an adj. *p* value < 0.05 in IR vs. sham comparison, *n* = 7 IR and *n* = 5 sham.

**Table 1 ncrna-09-00024-t001:** Serum Biochemistry. Analysis of serum collected 24 h after completion of sham or IR surgery. *****
*p*-value < 0.05, Student’s test; *n* = 5 sham, *n* = 7 IR.

		Sham Group	IR Group	
Test	Units	Mean ± SD	Mean ± SD	*p*-Value *
Urea	mg/dL	17.6 ± 3.9	96.5 ± 25.1	<0.001 *
Creatinine	mg/dL	0.32 ± 0.04	3.80 ± 1.42	0.001 *
Total Protein	g/dL	5.54 ± 0.21	5.47 ± 0.35	0.678
Albumin	g/dL	2.76 ± 0.11	2.63 ± 0.15	0.115
Glucose	mg/dL	175.4 ± 19.0	220.0 ± 38.1	0.025 *
Phosphorous	mg/dL	8.32 ± 0.60	12.84 ± 2.96	0.006 *
Calcium	mg/dL	9.62 ± 0.08	9.56 ± 0.34	0.654
Sodium	mmol/L	145.8 ± 0.8	140.4 ± 5.3	0.037 *
Potassium	mmol/L	5.24 ± 0.35	6.59 ± 1.81	0.099
Chloride	mmol/L	102.6 ± 1.1	104.0 ± 19.1	0.853
Bicarbonate	mmol/L	25.8 ± 0.5	18.0 ± 5.9	0.012 *
Anion Gap	mmol/L	22.6 ± 0.9	39.0 ± 27.5	0.166
AST(GOT)	U/L	266.2 ± 111.4	344.6 ± 60.7	0.205
ALT(GPT)	U/L	54.8 ± 16.2	81.7 ± 19.3	0.027 *
Alk. Ptase	U/L	93.0 ± 4.5	128.0 ± 19.1	0.002 *
Globulin	g/dL	2.78 ± 0.13	2.84 ± 0.21	0.544
Gamma-GT	U/L	0.2 ± 0.5	18.0 ± 3.3	<0.001 *
Triglycerides	mg/dL	38.2 ± 3.1	39.7 ± 5.4	0.551
Cholesterol	mg/dL	96.8 ± 14.4	87.3 ± 5.2	0.219

**Table 2 ncrna-09-00024-t002:** Differentially expressed miRs reported in ischemia-reperfusion injury models that share homology with humans, mice, and rats. Nineteen DE microRNAs relevant to kidney injury or damage with homology in human, mouse, and rat found within plasma, cortex, or urine samples collected 24 h after the ischemic event (adj. *p* value < 0.05).

Sample Type	miR and IR Studies References	*p* Value	adj. *p* Value	log2FC
**Plasma**	rno–miR–145–5p [12,13]	0.0057	0.0401	−0.7914
	rno–miR–150–5p [14,15,16,17]	0.0025	0.0221	−1.1176
	rno–miR–199a–3p [18,19]	0.0058	0.0401	−1.2610
	rno–miR–10a–3p [20]	<0.0001	<0.0001	2.2347
	rno–let–7a–5p [21,22]	0.0055	0.0381	0.3956
**Cortex**	hsa–miR–486–5p [23,24,25]	0.0045	0.0326	−0.9168
	rno–miR–30c–5p [26,27]	0.0018	0.0149	−0.7125
	rno–miR–132–3p [28,29]	<0.0001	<0.0001	2.2297
	rno–miR–146b–5p [28,30,31]	<0.0001	<0.0001	2.1612
	rno–miR–147 [31]	0.0022	0.0180	1.3324
	rno–miR–18a–5p [28,32]	<0.0001	0.0001	1.8245
	rno–miR–20a–5p [33,34,35,36]	0.0010	0.0098	0.8320
	rno–miR–21–5p [36,37,38]	<0.0001	<0.0001	1.3052
	rno–miR–27a–5p [39,40]	0.0001	0.0011	1.3213
	rno–miR–34c–5p [41]	<0.0001	<0.0001	1.2235
	rno–miR–98–5p [42,43]	0.0002	0.0026	0.5486
**Urine**	rno–miR–196b–5p [44]	0.0037	0.0363	−1.4190
	rno–miR–26a–5p [45]	0.0005	0.0084	−1.3099
	rno–miR–877 [46,47]	0.0031	0.0320	2.4805

## Data Availability

Complete results of profiling analysis are available in Appendix A.

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
