# Peer review of "Liquid Biopsies Poorly miRror Renal Ischemia-Reperfusion Injury"

_ncrna, 2023, doi:10.3390/ncrna9020024_

Round 1

Reviewer 1 Report

The submitted manuscript presents the analysis of expression of miRs in three types of samples derived from a rat model of ischemia/reperfusion induces acute kidney injury, i.e. renal cortex, blood plasma and urine. miRs profiling was followed by statistical analysis and Pubmed screening for miRs demonstrated to be differentially expressed both in the submitted and previous studies. The study design is appropriate regarding the aim. The results are clearly presented, properly interpreted and comprehensively discussed. The submission includes also supplementary materials that provide detailed results for miR profiles among the samples/experimental groups of animals. However, there are some minor issues/questions authors shall address in the revised version of the study before it could be considered for acceptance.

The table S2, sheet 1.1 provides data about mapping rate. Urinary samples are generally characterized by relatively low percentage of mapped reads and those for “Sham” group of animals are often below 20%. Is there any association between the % of unassigned reads of urine samples and the lack of any overlapping DE miRs in urine and cortex (despite the correlations presented in Fig. 2?)? What was the reason for lower % of mapped reads for these samples?

Lack of overlapping DE miRs in cortex and urine is interesting finding, are there any previous studies, including other disease/other animal models, documenting and discussing discrepancies between renal tissue and urine miRs expression profiles?

The study focuses on miRs, but the differences/similarities in expression profiles of miRs between the plasma -> cortex -> urine could be partially attributed to lncRNAs which can function as sponge for miRs. Did authors consider lncRNAs for further investigations using the rat model and/or available expression datasets?

There are some minor typos that need correction: “minutes” and “min” need unification (lines 315, 326, 328); missing spaces (line 249, 254 before the bracket; line 336, 349 between the value and unit), or misplaced dot before/after in-text citation (line 243).

Author Response

We appreciate the helpful and thoughtful comments provided in review of our manuscript.  Responses to specific comments are provided below.

1) The table S2, sheet 1.1 provides data about mapping rate. Urinary samples are generally characterized by relatively low percentage of mapped reads and those for “Sham” group of animals are often below 20%. Is there any association between the % of unassigned reads of urine samples and the lack of any overlapping DE miRs in urine and cortex (despite the correlations presented in Fig. 2?)? What was the reason for lower % of mapped reads for these samples?

Response: We cannot know for certain why the mapping rate in the sham urine was low.  It was relatively low in both sham and IR urine samples.  Our size-selective library preparation moves all miRNA-sized RNAs forward to sequencing and alignment is completed afterward. Given the high mapping rate in plasma, also a biofluid, we don’t think there is an issue with the library preparation, sequencing or alignment protocols.  Though we can not be certain, it is likely that a larger proportion of short RNAs in the urine samples are fragments of larger RNAs that would not have aligned with reference miRNAs.

2) Lack of overlapping DE miRs in cortex and urine is interesting finding, are there any previous studies, including other disease/other animal models, documenting and discussing discrepancies between renal tissue and urine miRs expression profiles?

Response: A number of studies have explored miRNA profiles between urine and renal tissue profiles (Kito et al. doi: 10.1155/2015/465479; Church et al. https://doi.org/10.1039/c4tx00051j). In the Kito et al. paper, the authors measured conserved miRNAs from human and rat tubular tissues, plasma, and urine to assess their potential as biomarkers of kidney disease severity. With this they found that the certain members of the miR-200 family were consistently elevated in all three sample types. Separately, in a human infant model of UUO, Papadopoulos et al. (2017; reference #21 in manuscript) analyzed both urine and kidney samples to understand molecular mechanisms involved in obstructive nephropathies. From their study, they gained insight into dysregulated miR expression profiles and highlighted that “there is no common regulation pattern in urine and kidney and in different species, even in the same pathological context”. In essence, our study recapitulates the uncertainty in using miRs as biomarkers of renal pathology due to unpredictability in both regulation and expression patterns during injury, though it is not an AKI model.

 Alternatively, the Church et al. paper established a relationship between miR-34c-3p levels in both glomeruli and urine following doxorubicin toxicity in Sprague Dawley rats, where after adjustment, miR-34c-3p was reduced when comparing glomerular tissue vs adjacent nonglomerular segments. These authors attributed the decrease renal miR-34c-3p to leakage into the urine (contributing to elevations of miR-34c-3p there).

We have updated the manuscript to reflect changes in the discussion section (lines 180-183) to include these papers as follows: “In a study evaluating the miR-200 family’s potential as a biomarker of kidney segment specific injury induced by doxorubicin, Church et al. (2014) attributed the decrease in renal miR-34c-3p to leakage into the urine, which contributing to elevations of urinary miR-34c-3p there.”

3) The study focuses on miRs, but the differences/similarities in expression profiles of miRs between the plasma -> cortex -> urine could be partially attributed to lncRNAs which can function as sponge for miRs. Did authors consider lncRNAs for further investigations using the rat model and/or available expression datasets? 

Response: We have not explored lncRNA this as a part of this study, because of the focus was on microRNAs as biomarkers, which are understood to be more stable than longer RNAs.  Because of the size-selective library preparation used for sequencing and the approach used, lncRNAs were not sequenced in this analysis.

4) There are some minor typos that need correction: “minutes” and “min” need unification (lines 315, 326, 328); missing spaces (line 249, 254 before the bracket; line 336, 349 between the value and unit), or misplaced dot before/after in-text citation (line 243).

Response: Thank you for pointing these typos out. We have corrected these errors in a re-proofing of the manuscript.

Reviewer 2 Report

In the manuscript entitled “Do liquid biopsies miRror renal ischemia reperfusion injury?” by Williams et al., the authors investigated and compared the miRNA profiles in renal cortex, urine and plasma samples collected from a rat model of ischemia/reperfusion-induced acute kidney injury.

The topic is within the aims and scope of Non-Coding RNA and particularly well-suited for the special issue entitled "Non-coding RNA in the USA: Latest Advances and Perspectives".

General comment:

As clearly stated in the introduction, the underlying hypothesis of the study was that “urinary and plasma miR profiles would correlate with changes in the renal cortex” in samples collected from a rat model of ischemia/reperfusion-induced acute kidney injury. Authors gathered experimental data to answer this question but decided not to give the response in the title of the manuscript. In my personal opinion, the current (intriguing and a little provocative) title might be more appropriate for a commentary or a review paper.

Though the hypothesis could not be confirmed by the experimental data, the manuscript shows deeper insights into the possibility for a miR to serve as a biomarker for renal failure with any solid clinical applicability.

Characterization of the isolated extracellular vesicles is completely missing and should be included. As conceived by the authors, this represents a major limitation of the study.

Minor issues:

1.                   Line 346: LV. Define abbreviation upon first use

2.                   Line 348: “approach [71]. and “. Delete .

3.                   The progression of the injury in the rat model of ischemia/reperfusion-induced AKI has been previously evaluated (doi: 10.3109/0886022X.2012.725292). How these data relate with the sampling time (22 hours after injury) you used in the current study?

4.                   Table 1: total bilirubin in sham controls is mg/dL 0.00  ±  0.00, while is should be detectable in wild type Sprague Dawley rats. Similarly, the level in IR group should be considered within the normal range, raising concerns about the meaning of the difference observed between the two groups. Could this be related to the sensitivity of the assay used?

5.                   Table 1: glucose levels in both groups seems to be elevated. Dis you by any chance evaluated urine glucose content? Can you comment on these values?

Author Response

We appreciate the helpful and thoughtful comments provided in review of our manuscript.  Responses to specific comments are provided below.

1) As clearly stated in the introduction, the underlying hypothesis of the study was that “urinary and plasma miR profiles would correlate with changes in the renal cortex” in samples collected from a rat model of ischemia/reperfusion-induced acute kidney injury. Authors gathered experimental data to answer this question but decided not to give the response in the title of the manuscript. In my personal opinion, the current (intriguing and a little provocative) title might be more appropriate for a commentary or a review paper. 

Response: Thank you for this comment.  Though this was the research questions addressed in our study, we see how it may be more beneficial to incorporate the answer.  The title has been updated accordingly.

2) Though the hypothesis could not be confirmed by the experimental data, the manuscript shows deeper insights into the possibility for a miR to serve as a biomarker for renal failure with any solid clinical applicability.  

Response: We agree. 

3) Characterization of the isolated extracellular vesicles is completely missing and should be included. As conceived by the authors, this represents a major limitation of the study.

Response: We appreciate this comment.  We intentionally did not want to focus on specific extracellular vesicle types, and for that reason took all of the vesicles from the serum, which would be the most clinically applicable approach.  While we did not characterize the urinary vesicles that were isolated, the centrifugation approach should have captured exosome microvesicles, based on published reports.  There are many ways to characterize vesicles, which will require a much more extensive set of analyses, and large sample quantity, including characterization by size and numerous protein surface markers. This would require separate RNA sequencing for each subcategory of vesicle as well. This study provides a strong rationale to complete this set of studies in the future, allowing us to understand which type of vesicles contribute specific miRNAs.

Minor issues:

4)  Line 346: LV. Define abbreviation upon first use

This has been corrected.

5)  Line 348: “approach [71]. and “. Delete .

This has been corrected.

6)  The progression of the injury in the rat model of ischemia/reperfusion-induced AKI has been previously evaluated (doi: 10.3109/0886022X.2012.725292). How these data relate with the sampling time (22 hours after injury) you used in the current study?

Response:  The extent of damage in the paper referenced about was more severe than that in our paper, as ischemia was induced for at least 60 minutes in each kidney.  Our 30 minutes of ischemia is a better model for AKI-induced CKD, whereas 60+ minutes of ischemia results in AKI-induced renal failure.

7) Table 1: total bilirubin in sham controls is mg/dL 0.00  ±  0.00, while is should be detectable in wild type Sprague Dawley rats. Similarly, the level in IR group should be considered within the normal range, raising concerns about the meaning of the difference observed between the two groups. Could this be related to the sensitivity of the assay used? 

Response:  Thank you for raising this issue.  Indeed, this could be related to the sensitivity of the assay for this specific metabolite even though this analysis was completed at a clinical laboratory.  Circulating bilirubin is higher in humans than rats.  As was noted, bilirubin was undetected in the sham animals, but was also undetected in 2 of 7 of the IR animals.  Given this measurement was out of range, and it is not central to the goals, hypothesis, or conclusions of the paper, we removed bilirubin measurements from the table.

8) Table 1: glucose levels in both groups seems to be elevated. Dis you by any chance evaluated urine glucose content? Can you comment on these values?

Response: We did not measure urine glucose, but also found changes in the serum panel to be an interesting finding and wanted to include this data.  Unfortunately, we are unable to comment further about urinary glucose in this model.